# The Role of Neoadjuvant Chemotherapy in Repeat Local Treatment of Recurrent Colorectal Liver Metastases: A Systematic Review and Meta-Analysis

**DOI:** 10.3390/cancers13030378

**Published:** 2021-01-20

**Authors:** Madelon Dijkstra, Sanne Nieuwenhuizen, Robbert S. Puijk, Bart Geboers, Florentine E. F. Timmer, Evelien A. C. Schouten, Hester J. Scheffer, Jan J. J. de Vries, Johannes C. F. Ket, Kathelijn S. Versteeg, Martijn R. Meijerink, M. Petrousjka van den Tol

**Affiliations:** 1Department of Radiology and Nuclear Medicine, Amsterdam University Medical Center, VU Medical Center Amsterdam, 1081 HV Amsterdam, The Netherlands; s.nieuwenhuizen1@amsterdamumc.nl (S.N.); r.puijk@amsterdamumc.nl (R.S.P.); b.geboers@amsterdamumc.nl (B.G.); f.timmer1@amsterdamumc.nl (F.E.F.T.); e.schouten@amsterdamumc.nl (E.A.C.S.); hj.scheffer@amsterdamumc.nl (H.J.S.); j.devries1@amsterdamumc.nl (J.J.J.d.V.); mr.meijerink@amsterdamumc.nl (M.R.M.); 2Medical Library, Vrije Universiteit Amsterdam, 1081 HV Amsterdam, The Netherlands; h.ket@vu.nl; 3Department of Medical Oncology, Amsterdam University Medical Center, VU Medical Center Amsterdam Cancer Center Amsterdam, 1081 HV Amsterdam, The Netherlands; k.versteeg@amsterdamumc.nl; 4Department of Surgery, Amsterdam University Medical Center, location VU Medical Center Amsterdam, 1081 HV Amsterdam, The Netherlands; mp.vandentol@amsterdamumc.nl

**Keywords:** repeat local treatment, thermal ablation, partial hepatectomy, (neoadjuvant) chemotherapy, recurrent colorectal liver metastases (CRLM)

## Abstract

**Simple Summary:**

Up to 85% of patients with colorectal liver metastases develop distant intrahepatic recurrence after curative intent local treatment. (Inter)national guidelines and scientific societies consider repeat local treatment, comprising repeat thermal ablation and/or repeat resection, the standard of care to treat recurrent new colorectal liver metastases. This systematic review and meta-analysis assessed the potential additive value of neoadjuvant chemotherapy before repeat local treatment. The addition of neoadjuvant chemotherapy prior to repeat local treatment was suggested by merely all authors, though supporting evidence is lacking. The results do not substantiate the routine use of neoadjuvant chemotherapy. We are currently constructing a phase III randomized controlled trial directly comparing upfront repeat local treatment with neoadjuvant chemotherapy followed by repeat local treatment (COLLISION RELAPSE trial).

**Abstract:**

The additive value of neoadjuvant chemotherapy (NAC) prior to repeat local treatment of patients with recurrent colorectal liver metastases (CRLM) is unclear. A systematic search was performed in PubMed, Embase, Web of Science, and an additional search in Google Scholar to find articles comparing repeat local treatment by partial hepatectomy and/or thermal ablation with versus without NAC. The search included randomized trials and comparative observational studies with univariate/multivariate analysis and/or matching as well as (inter)national guidelines assessed using the AGREE II instrument. The search identified 21,832 records; 172 were selected for full-text review; 20 were included: 20 comparative observational studies were evaluated. Literature to evaluate the additive value of NAC prior to repeat local treatment was limited. Outcomes of NAC were often reported as subgroup analyses and reporting of results was frequently unclear. Assessment of the seven studies that qualified for inclusion in the meta-analysis showed conflicting results. Only one study reported a significant difference in overall survival (OS) favoring NAC prior to repeat local treatment. However, further analysis revealed a high risk for residual bias, because only a selected group of chemo-responders qualified for repeat local treatment, disregarding the non-responders who did not qualify. All guidelines that specifically mention recurrent disease (3/3) recommend repeat local treatment; none provide recommendations about the role of NAC. The inconclusive findings of this meta-analysis do not support recommendations to routinely favor NAC prior to repeat local treatment. This emphasizes the need to investigate the additive value of NAC prior to repeat local treatment of patients with recurrent CRLM in a future phase 3 randomized controlled trial (RCT).

## 1. Introduction

Colorectal cancer (CRC) is the second most common cancer type in women and the third most common in men; it represents about 10% of the annual global cancer incidence [1]. The prognosis of CRC patients largely depends on the presence of distant metastasis, the liver being the most frequently involved organ. Up to 50% of patients develop colorectal liver metastases (CRLM) during the course of disease [2,3,4,5,6,7]. If left untreated, five-year overall survival (OS) is <3% and when treated with palliative chemotherapy alone this improves to approximately 11% [3,8,9,10]. 

One-fifth of patients who develop CRLM are eligible for curative intent local treatment options, such as partial hepatectomy or thermal ablation (radiofrequency ablation, RFA; microwave ablation; MWA) [3,11,12,13,14,15,16,17]. The five-year OS for upfront resectable and/or ablatable disease nowadays reaches 44–58% [18,19,20,21,22,23,24,25,26,27,28] and even up to 33% for an increasing number of patients with initially unresectable and unablatable disease who are successfully downstaged after induction chemotherapy [12].

Although leaving room for debate, the absence of a survival benefit of perioperative FOLFOX4 chemotherapy in the EORTC 40983 trial by Nordlinger and colleagues [29] has put an end to the routine use of perioperative chemotherapy in case of resectable and/or ablatable disease. The European COLLISION trial group proposed two exceptions to that rule: (1) if tumor eradication requires major hepatectomy and the chemo-regimen is likely to reduce procedural risk or (2) if poor tumor biology is suggested by the appearance of (new) liver metastases within 6 months following primary tumor diagnosis [30]. More recently, the JCOG 0603 trial suggested that postoperative chemotherapy with mFOLFOX6 improves disease-free survival (DFS) but worsens overall survival (OS) over local treatment alone [31]. As a result, the authors concluded that adjuvant chemotherapy is not indicated.

Both resection and ablation offer complete local tumor eradication in the vast majority of cases. However, in 64–85% of patients distant intrahepatic recurrence develops [32,33,34]. Several retrospective comparative series, using either propensity-score matching or multivariate analysis, revealed a superior DFS and OS for repeat local treatment (+/- peri-procedural systemic chemotherapy) over palliative chemotherapy alone with a high rate of long-term survivors [35,36,37]. Therefore, most consider thermal ablation or partial hepatectomy the standard of care to treat recurrent new CRLM. Given the poorer prognosis and presumed worse tumor biology of patients with recurring disease, neoadjuvant chemotherapy (NAC) prior to repeat local treatment has been suggested to prolong survival and to select responders who will benefit most from local treatment [38,39,40,41]. In the absence of prospective randomized controlled trials, these claims are being challenged by the negative results from the EORTC 40983 and JCOG 0603 series. Here, (peri)operative chemotherapy was administered concomitant with the first local treatment, leading to well-known risks associated with liver surgery following repeated cycles of oxaliplatin (blue liver syndrome or sinusoidal obstruction syndrome) and irinotecan (yellow liver/liver steatosis), systemic toxicity and added direct costs [13,29,31,42,43]. These arguments question the added value of NAC prior to repeat local treatment.

The aim of this systematic review and meta-analysis was to assess the role of NAC prior to repeat local treatment in case of recurrent new CRLM.

## 2. Materials and Methods 

This systematic review and meta-analysis is reported in accordance with the Preferred Reporting Items for Systematic Reviews and Meta-Analyses (PRISMA) statement and PICO (patients, interventions, comparisons, outcomes) protocol [44]. 

### 2.1. Search Strategies

PubMed, Embase.com, and Clarivate Analytics/Web of Science Core Collection were systematically searched from inception up to 29 October 2020 (by MD and JCFK) for publications reporting on the outcomes of NAC before local treatment of recurrent CRLM. Google Scholar was searched (by MD) on 2 November 2020 for additional references. The following PICO question was used for the search strategy and inclusion criteria: *p* (population): patients with recurrent CRLM; (I) intervention: neoadjuvant chemotherapy (NAC) before repeat local treatment (repeat ablation and/or resection); (C) comparison: repeat local treatment alone; (O) outcome: the critical endpoint was overall survival (OS), important endpoints were disease-free survival (DFS), complications, quality of life (QoL), and cost-effectiveness. In the search we used both keywords or free text terms from the PICO formulated question for ‘colorectal liver metastases’ and ‘repeat’ and ‘(neoadjuvant) chemotherapy’. The full search strategy, supported by a medical information specialist (JCFK), can be found in Appendix A. The literature search excluded case reports, conference abstracts (in Embase) and animal studies, no limits on date or language were applied. 

### 2.2. Study Selection

One review author (MD) assessed the titles and abstracts retrieved by the literature search. Studies that appeared relevant were subjected to a full-text evaluation; reviews were excluded. References of potentially relevant studies were also evaluated and if eligible subjected to full-text evaluation. Included articles were original articles that directly compared NAC plus repeat local treatment to upfront local treatment alone (prospective randomized controlled trials or non-randomized comparative series with case matching or multivariate analysis), provided that at least one critical or important endpoint was reported. Exclusion criteria were articles that did not specifically assess recurring disease, articles that merely assessed local tumor progression at the treatment-site, articles without a comparator or with an incorrect comparator, series where chemotherapy was administered as adjuvant therapy after repeat local treatment and articles where (part of) the chemotherapy regimen was considered off-label such as a trans-arterial administration or experimental drugs or combinations of drugs.

### 2.3. Data Extraction

One review author (MD) extracted the following variables from each included study: author, year, years of inclusion, study design, number of patients with repeat local treatment, age, chemotherapy regimen specifics, local treatment procedures, mortality and morbidity rates, recommendations, outcomes, and conclusions. The data from the relevant articles were extracted from texts, tables, and figures. The extracted data were checked by a second author (MRM) and disagreements were resolved by consensus; if no consensus was reached, a third author was planned to decide. The results were presented as reported in the articles, with recalculations of the extracted data if necessary to match the format of tables and figures of this study. 

### 2.4. Data Analysis

A qualitative systematic analysis was performed by tabulation of the results with the abovementioned variables and an assessment was made whether the studies were sufficiently comparable to perform a meta-analysis. To account for statistical heterogeneity a random effects model was chosen; results were presented in forest plots. For time-to-event outcomes (OS and DFS), the generic inverse variance method was used and the hazard ratios (HR) from univariate and multivariate analysis were utilized. For dichotomic results (1-, 3-, and 5-year OS, QoL, and complications), the Mantel–Haenszel method was used to calculate risk ratios (RR). Review Manager 5.3 was used to perform the meta-analysis.

### 2.5. Guidelines

(Inter)national guidelines were searched on websites of (inter)national guideline organizations and scientific societies and reviewed using the AGREE II instrument (Appendix B) by two reviewers (MD, MRM).

## 3. Results

The literature search identified 31,998 articles. After removal of duplicates a total of 21,832 articles were screened. Based on title and abstract 172 articles were selected for full-text review. The following reasons for exclusion were found: no comparator (*n* = 111) or wrong comparator (*n* = 41). Twenty articles met the inclusion criteria for qualitative synthesis by tabulation. In addition, seven articles used univariate analysis or sufficiently corrected for potential confounders using matching and/or multivariate analysis to allow inclusion in the meta-analysis (quantitative synthesis). Results regarding the comparison of NAC plus repeat local treatment versus upfront repeat local treatment alone were mentioned as incidental result in the majority of the included articles. The search results are presented in Figure 1: A flow diagram of the systematic search and selection according to PRISMA [44].

### 3.1. Study Characteristics 

Table 1 shows the study characteristics of the 20 included articles. The year of publication ranged from 2003 to 2019 and the years of inclusion from 1974 to 2016. All studies had a retrospective design. The total cumulative number of patients treated with repeat local treatment extracted from the included studies was 2366 (range 17–447 per study; 1108 NAC + rLT (repeat local treatment) versus 807 rLT alone). Data from patients in the included articles that did not fulfill inclusion criteria were excluded from the table. Mean age of patients ranged 56–66 (individual patients: 24–95 years).

### 3.2. Treatment Characteristics

Table 2 defines reported treatments for recurrent CRLM, type of chemotherapy regimens and local treatment(s) used. It also shows the overall mortality and morbidity rates following repeat local treatment. The subgroups’ percentages on chemotherapy before local treatment differed strongly per study and this was predominantly based on local expertise, most often determined by multidisciplinary tumor board evaluations. Specific motives when NAC plus repeat local treatment should be favored over upfront repeat local treatment were not provided. 5-fluorouracil (5-FU or F), oxaliplatin (OX), and irinotecan (IRI) were frequently used as chemotherapeutic agents. Local treatment procedures were divided into thermal ablations and/or non-anatomical metastasectomy, segmentectomy, minor (<3 segments: metastasectomy, (bi)segmentectomy, caudate resection) and major (≥3 segments: trisectionectomy, (extended) hemihepatectomy) hepatectomies without providing further definitions.

### 3.3. Level of Evidence

The level of evidence to reliably assess the additive value of NAC was very low [65]. Direct comparisons of NAC plus repeat local treatment versus repeat local treatment alone were merely mentioned as subgroup analyses in all included series. In 13/20 articles, the specific statistical method used to compare the groups was either unsatisfactory or could not be deduced. 

### 3.4. Overall Survival (Critical Endpoint) and Disease-Free Survival (Important Endpoint)

Seven studies [46,48,59,60,61,63,64] were compatible for meta-analysis, presented in Figure 2, Figure 3, Figure 4 and Figure 5, respectively reporting OS and the 1-, 3-, and 5-year OS (Table 3). Viganò et al. reported HR = 0.529 (0.299–0.934; *p*-value = 0.028) for OS. Generic inverse variance with random effects analysis model showed HR = 0.76 (0.48–1.19; *p*-value = 0.22) for OS. Mantel–Haenszel method with random effect analysis model showed a one-year OS RR of 0.87 (0.33–2.29; *p*-value = 0.78), a three-year OS RR of 1.41 (0.73–2.73; *p*-value = 0.31), and a five-year OS RR of 0.91 (0.78–1.07; *p*-value = 0.25). Two series did not report a difference in OS [45,57], five suggested a trend towards improved survival in selected patient groups treated with NAC [51,54,55,56,58] without defining eligibility criteria for receiving NAC (Table 3 and Table 4).

Adair et al. and Heise et al. reported no difference in disease-free survival between upfront repeat local treatment and NAC followed by repeat local treatment (*p*-values 0.250–0.483) [45,52] (Table 3).

### 3.5. Complications, Quality of Life, and Cost-Effectiveness (Important Endpoints)

Overall mortality rates were below 5% and average complication rate was 24.8% with an average re-operation rate of 6.6% to resolve iatrogenic complications of the surgical or interventional procedure (Table 2). Stratification of mortality and complication rates for patients receiving NAC followed by repeat local treatment versus upfront repeat local treatment was not reported.

Butte et al. and Andreou and colleagues reported no difference in effectiveness of surgical results [47,49] and the future liver remnant volume did not differ between patients with and without chemotherapy according to Valdimarsson and colleagues [62]. However, Hallet and colleagues found a more congested and friable liver parenchyma after NAC before repeat local treatment, which may increase the risk of bleeding during surgery [50] (Table 3 and Table 4).

Although Heise and colleagues mention no difference in QoL between the initial local treatment and repeat local treatment [52], a formal statistical analysis between patients receiving NAC followed by repeat local treatment and upfront repeat local treatment with regards to quality of life or cost-effectiveness was not described in any of the articles.

### 3.6. Guidelines 

A total of 15 guidelines were selected for full-text analysis. Twelve guidelines did not report on the treatment of recurrent CRLM [66,67,68,69,70,71,72,73,74,75]. This resulted in three eligible guidelines. The UK National Institute for Health and Care Excellence (NICE) guideline recommends that recurrent CRLM should be treated with repeat partial hepatectomy or thermal ablation [76,77,78]. When unfit or ineligible for these procedures, systemic chemotherapy should be preferred. The Japanese Society for Cancer of the Colon and Rectum (JSCCR) also states that repeat hepatectomy should be taken into consideration, with similar (contra-)indications as for initial hepatectomy [79]. The Dutch Comprehensive Cancer Centre (IKNL) guideline reported repeat surgical treatment or thermal ablation as optional for carefully selected patients [80]. No recommendations with regards to NAC prior to repeat local treatment were found.

## 4. Discussion

Contradictory to claims advising the use of NAC before repeat local treatment in all or in selected patients with recurrent liver-confined CRLM, evidence to support these is weak [45,46,47,48,51,53,54,55,56,57,58,59,60,61,64]. Only Viganò and colleagues (LiverMetStudy) reported a superior OS favoring the use of NAC before repeat local treatment (HR = 0.529; 95%CI 0.299–0.934; *p* = 0.028) [63]. Parameters such as early disease recurrence, a certain number of or rapidly growing metastases, a high clinical risk score (CRS by Fong and colleagues, modified scores by Brudvik et al. and, specifically for RFA procedures, by Sofocleous et al. and by Shady et al.), high carcinoembryonic antigen (CEA) levels and specific molecular genome mutations or consensus molecular subtypes are well-known prognosticators [33,48,54,56,63,81,82,83]. However, they currently cannot and should not be used as predictive biomarkers to exclude patients from receiving repeat local treatment or to routinely allocate them to receiving NAC first. Two conceivable exemptions to that rule-of-thumb are (a) tumor genome microsatellite instability (MSI), where checkpoint blockade has shown durable responses in high rates of patients, and (b) if chemotherapy is likely to reduce the risks of a certain procedure, both patient groups should preferably be treated with chemotherapy first followed by repeat local treatment [30]. In the latter example, the systemic regimen should be referred to as induction chemotherapy, contrary to neoadjuvant chemotherapy.

Although evidence to support repeat local treatment also lacks substantiation from randomized controlled trials, it is better-established, more widely adopted and currently not considered to be in equipoise with treating recurrent liver-confined CRLM patients with chemotherapy alone [35,36,37]. The inconclusive findings of the role of NAC in repeat local treatment of recurrent CRLM seems to be in line with the conflicting results of studies assessing the role of (neo)adjuvant chemotherapy concomitant with the initial local treatment [12,29,31,39,84]. 

This systematic review and meta-analysis has several limitations. An important limitation is the long study duration of the included studies in the analysis. Though fluorouracil (5-FU) based chemotherapy combined with oxaliplatin and/or irinotecan became the standard first-line therapy in the early 2000s and no series prior to 2003 were included, some included series covered patients treated prior to 2000, leading to population bias [85]. In a number of studies, the specific chemotherapy regimen was not clearly reported or varied over time. Moreover, biological agents were not routinely administered. Nonetheless, no chronological change in hazard ratios between the two groups assessed—upfront repeat local treatment versus NAC followed by repeat local treatment—was identified. Due to the absence of randomized controlled trials, the inclusion of merely comparative observational studies led to lower levels of evidence, selection bias and a high risk of publication bias. The selection of patients for NAC was not preceded by protocol. Although partly accounted for in multivariate analyses, residual confounding patient and disease characteristics that could affect the reason to opt for NAC are age, comorbidities, synchronous disease, positive lymph node status (primary tumor), disease-free interval <12 months, multiple metastases, largest liver metastasis >5 cm, CEA level > 200, RAS- and BRAF-mutations, MSI/MSS and extrahepatic disease [30,81,86]. In addition, the colon life nomogram of Pietrantonio and colleagues proposed a performance status ≤2, primary tumor resection, lactate dehydrogenase (LDH) and peritoneal involvement to predict the probability of death <12 weeks in recurrent CRLM [87,88]. Results regarding the comparison of NAC plus repeat local treatment versus upfront repeat local treatment alone were mentioned as results from subgroup analysis in the majority of included articles and limited sample sizes were found. In other studies, differentiation between NAC and induction chemotherapy was unclear in disease either (a) initially unsuitable for repeat local treatment or (b) in disease where downsizing chemotherapy could potentially reduce procedural risks. Especially this presumed risk reduction is poorly defined and as a result patients from these series may overlap with patients eligible for upfront local retreatment with or without NAC. 

The pooled results from this meta-analysis together with the negative results from the EORTC 40983 and JCOG 0603 trials for NAC prior to the initial local treatment, and the absence of prospective randomized controlled studies for recurrent CRLM disqualify the validation to routinely advocate NAC prior to repeat local treatment. The lack of clear recommendations regarding recurrent CRLM in (inter)national guideline organizations and scientific societies and the well-known risks associated with local treatment following repeated cycles of chemotherapy further challenge the substantiation to use NAC, even in subgroups of locally retreatable patients. These results underline the importance of comparative research analyzing the added value of NAC prior to repeat local treatment of patients with recurrent CRLM. We are currently constructing a phase III randomized controlled trial (RCT) directly comparing upfront repeat local treatment with NAC followed by repeat local treatment (COLLISION RELAPSE trial).

## 5. Conclusions

To conclude, the findings of this meta-analysis do not support recommendations to routinely favor NAC prior to repeat local treatment as means to select good candidates or to control rapidly progressive disease. This emphasizes the necessity to investigate the additional value of NAC prior to repeat local treatment of patients with recurrent CRLM in a future phase 3 RCT. 

## Figures and Tables

**Figure 1 cancers-13-00378-f001:**
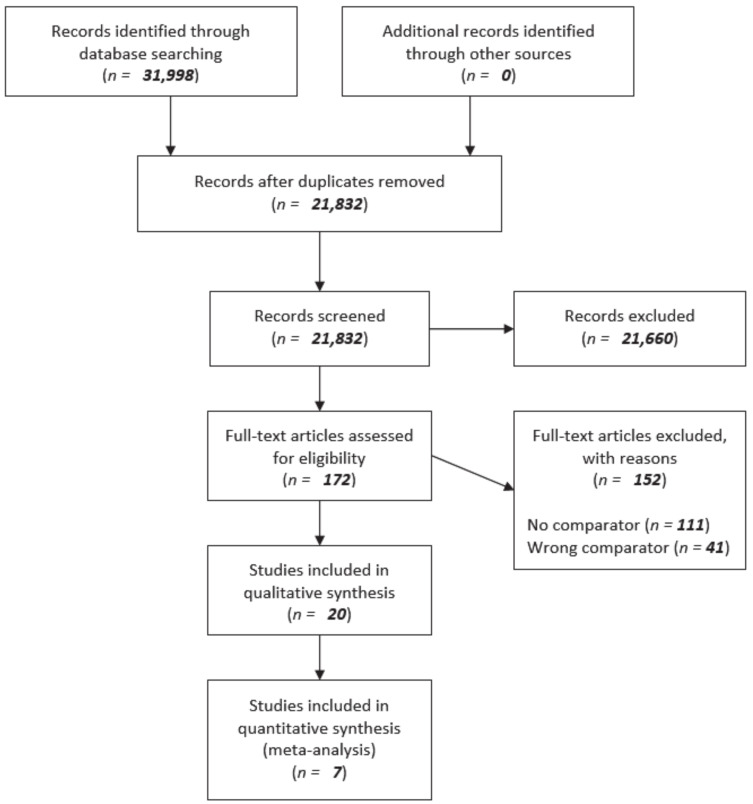
Flow diagram of systematic search and selection according to PRISMA [44].

**Figure 2 cancers-13-00378-f002:**
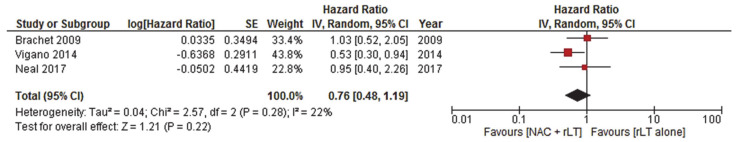
NAC + rLT versus rLT alone: OS.

**Figure 3 cancers-13-00378-f003:**
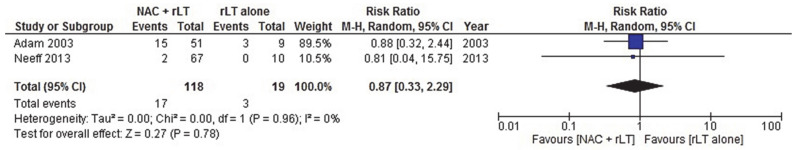
NAC + rLT versus rLT alone: one-year OS.

**Figure 4 cancers-13-00378-f004:**
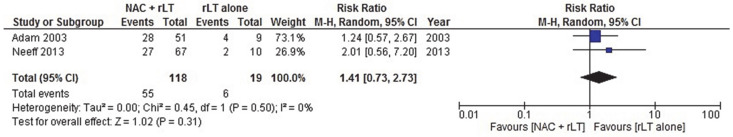
NAC + rLT versus rLT alone: three-year OS.

**Figure 5 cancers-13-00378-f005:**
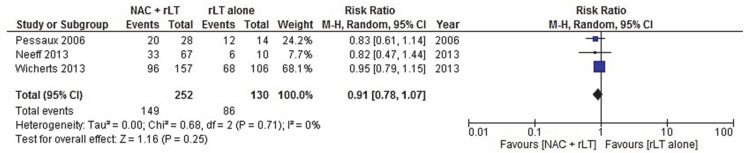
NAC + rLT versus rLT alone: five-year OS.

**Table 1 cancers-13-00378-t001:** Characteristics of studies reporting on patients with repeat local treatment for recurrent colorectal liver metastases including years of inclusion, study design, number of patients, and age.

Author	Year	Years of Inclusion	Number of Patients with Repeat Local Treatment	Median Age in Years (Range)	Mean Age in Years ± SD
Adair [45]	2012	1993–2010	195	63 (24–85)	NR
Adam [46]	2003	1984–2000	139 (2nd) * 60 (3rd) *	NR (32–78) NR (33–86)	56 ± 10 56 ± 10
Andreou [47]	2011	1993–2009	43	55 (32–74)	NR
Brachet [48]	2009	1992–2007	62 (2nd) * 15 (3rd) * 2 (4th) *	NR NR NR	62.2 ± 9.7 57.9 ± 10.8 54.5 ± 10.6
Butte [49]	2015	1994–2004	159 (PST)	59 (31–81)	58.1 ± 10.9
Hallet [50]	2016	2006–2013	447	61.4 (NR)	NR
Hashimoto [51]	2016	2000–2012	17	60 (40–80)	NR
Heise [52]	2017	2010–2016	38	59.1 (NR)	NR ± 12.1
Homayounfar [53]	2013	2001–2011	52	62.8 (NR)	NR
Imai [54]	2018	2000–2016	54	64 (25–94) ^a^	63.1 ± 11.0 ^a^
Imai [55]	2019	1992–2012	29	57 (31–82) ^a^	55.8 ± 9.9 ^a^
Ishiguro [56]	2006	1985–2004	111	NR	59 ± NR
Kishi [57]	2019	2000–2015	115	59 (28–86)	NR
Matsuoka [58]	2019	1974–2016	59	NR (25–95) ^a^	66 ± 11.2 ^a^
Neal [59]	2017	2001–2010	71	64 (26–85) ^a^	63.4 ± NR ^a^
Neeff [60]	2013	1999–2011	77	NR	NR
Pessaux [61]	2006	1992–2002	42	NR (34–80)	63.5 ± NR
Valdimarsson [62]	2019	2005–2015	82	64 (NR)	NR
Viganò [63]	2014	1998–2009	234	NR	NR
Wicherts [64]	2013	1990–2010	263	NR	57 ± 11

Note: SD = standard deviation; NR = not reported; PST = potential salvage therapy (re-resection of all recurrent disease); * = Rank from total number of hepatectomies; ^a^ = age of total population.

**Table 2 cancers-13-00378-t002:** Subgroups of treatment for recurrent CRLM, type of procedures, mortality, and morbidity rates.

Author	Chemotherapy + rLT, *n* (%)	Chemotherapeutic Agents (%)	rLT alone, *n* (%)	rLT Procedure (%)	Overall Mortality and Morbidity Rate (%)
Adair [45]	52 (26.7%)	FUFOL (28.8%) FOLFOX (34.6%) FUFOL + irinotecan (9.6%) Capecitabine (3.8%) Oxaliplatin (23.1%)	143 (73.3%)	Metastasectomy (70.8%) Segmentectomy (10.3%) Hemihepatectomy (12.2%) Trisectionectomy (4.6%) Caudate resection (2.1%)	30-day mortality 1.5% 30-day morbidity 20% (4.6% relaparotomy)
Adam [46]	2nd *: 127 (91%) 3rd *: 51 (85%)	FOLFOX (NR)	2nd *: 12 (9%) 3rd *: 9 (15%)	2nd * - Liver resection <3 segments (76%) - Major hepatectomy (24%) 3rd * - Liver resection <3 segments (41%) - Major hepatectomy (59%)	Mortality <2 months - 2nd *: 2.5% - 3rd *: 0% Morbidity - 2nd *: 23% (8% reoperation) - 3rd *: 25% (3% reoperation)
Andreou [47]	19 (44%)	NR	24 (56%)	Minor resections <3 segments (88%) Major liver resection ≥ 3 segments (12%)	30-day mortality 0% 90-day mortality 0% Morbidity 12% (0% required intervention)
Brachet [48]	2nd *: 38 (61.3%) 3rd *: 6 (40%) 4th *: 0 (0%)	2nd * - FUFOL (55.6%) - FOLFOX (22.2%) - FOLFIRI (19.4%) - Others (2.8%) 3rd * - FUFOL (50%) - FOLFOX (50%) - FOLFIRI (0%) - Others (0%) 4th *: No chemotherapy	2nd *: 24 (38.7%) 3rd *: 9 (60%) 4th *: 2 (100%)	2nd * - Hemihepatectomy (24.2%) - Segmentectomy (43.6%) - Nonanatomic (32.2%) 3rd * - Hemihepatectomy (6.7%) - Segmentectomy (46.6%) - Nonanatomic (46.7%) 4th * - Hemihepatectomy (50%) - Nonanatomic (50%)	Mortality <30 days - 2nd *: 0% - 3rd *: 0% - 4th *: 0% Morbidity <30 days (3.8% reoperation) - 2nd *: 17.7% - 3rd *: 26.7% - 4th *: 50%
Butte [49]	47 (30%)	NR	112 (70%)	Minor hepatectomy <hemi-liver (41%) Major hepatectomy: hemi-, central or extended (59%)	NR
Hallet [50]	310 (69.4%)	NR	137 (30.6%)	NR	Mortality <30 days 1.3% Morbidity <30 days 28.9% (8.1% re-intervention)
Hashimoto [51]	4 (24%)	Oxaliplatin (NR) or irinotecan (NR)	13 (76%)	Minor hepatectomy: wedge, segmental or sectional (88.2%) Major hepatectomy ≥ 3 segments (11.8%)	NR Morbidity 17.7%
Heise [52]	36 (95%)	NR	2 (5%)	Minor hepatectomy (76%) Major hepatectomy >hemi-liver (24%)	NR Morbidity 3%
Homayounfar [53]	10 (19%)	5FU (9%) 5FU + oxaliplatin (36%) 5FU + irinotecan (55%) Additional cetuximab (19%) Additional bevacizumab (34%)	42 (81%)	Surgical exploration only (5.8%) Surgical exploration + RFA liver (7.7%) Non-anatomic liver resection (38%) Bisegmentectomy (3.8%) Bisegmentectomy + nonanatomic liver resection (1.9%) Hemihepatectomy (5.8%) Trisectorectomy (1.9%) Rectal resection (5.8%) Rectal extirpation + nonanatomic liver resection (1.9%) Others (26.9%)	Mortality 0% Morbidity 26%
Imai, 2018 [54]	28 (51.9%)	Oxaliplatin-based (14.3%) Oxaliplatin-based + biologic agents (17.9%) Irinotecan-based (7.1%) Irinotecan-based + biologic agents (42.9%) Oxaliplatin and irinotecan-based + biologic agents (3.6%) Others (14.3%)	26 (48.1%)	Hepatectomy (38.9%) Hepatectomy + RFA (5.6%) Hepatectomy + resection of peritoneal metastasis (1.9%) RFA for liver metastasis (14.8%) RFA for liver + lung metastasis (1.9%) Others (37.0%)	NR
Imai, 2019 [55]	28 (73.7%)	NR	10 (26.3%)	NR	NR
Ishiguro [56]	NR	NR	NR	Minor resection (89.2%) Hemihepatectomy (5.4%) Extended hemihepatectomy (4.5%) Central bisectionectomy (0.9%)	Mortality 0% Morbidity 14%
Kishi [57]	6 (5.2%)	Oxaliplatin-based (NR) Irinotecan-based (NR) 5-FU with leucovorin (NR) Tegafur, Gimeracil, Oteracil Potassium (NR)	109 (94.8%)	NR	Mortality 0.9% Morbidity 27%
Matsuoka [58]	55 (93%)	NR	4 (7%)	Sectionectomy (26%) Segmentectomy (10%) Partial resection (64%)	Mortality 5% Morbidity 39%
Neal [59]	8 (11.3%)	NR	63 (88.7%)	Anatomical resection (19.7%) Major hepatectomy ≥ 3 segments (16.9%) NR	Mortality <90 days Morbidity <90 days 21.1%
Neeff [60]	67 (87%)	All 5-FU based	10 (13%)	Atypical/wedge (39.1%) Segmental (23.9%) Hemihepatectomy (17.4%) Extended hemihepatectomy (17.4%) Central resection (2.2%)	Mortality 3.3% Morbidity 53.3% (12% operative revisions)
Pessaux [61]	28 (66.7%)	NR	14 (33.3%)	Anatomic hepatectomy (66.7%) Non-anatomic hepatectomy (33.3%)	Mortality <30 days 0% Morbidity <30 days 14.3%
Valdimarsson [62]	37 (45%)	All oxaliplatin based	45 (55%)	Major liver procedure ≥ 3 segments 19% Minor liver procedure 81%	NR Morbidity 18%
Viganò [63]	NR	Oxaliplatin-based (NR) Irinotecan-based (NR) Associated cetuximab (NR) Associated bevacizumab (NR)	NR	Anatomic resection (NR) Non-anatomic resection (NR) Associated intraoperative RFA	NR
Wicherts [64]	157 (60.7%)	Last line regimen - 5-FU, LV (14.2%) - 5-FU, LV, oxaliplatin (41.9%) - 5-FU, LV, irinotecan (14.2%) - Others (29.7%) Biological agents last line - Cetuximab (7.6%) - Bevacizumab (3.8%)	106 (40.3%)	Major resection ≥3 segments (17.0%) Anatomical (27.9%) Non-anatomical (52.1%) Both anatomical and non-anatomical (20.0%)	90-day mortality 2.4% Morbidity 34.4%

Note: rLT = repeat local treatment; FUFOL = folinic acid and 5-fluorouracil; FOLFOX = folinic acid, 5-fluorouracil and oxaliplatin; NR = not reported; * = Rank from total number of hepatectomies; FOLFIRI = folinic acid, 5-fluorouracil and irinotecan; 5-FU = 5-fluorouracil; RFA = radiofrequency ablation.

**Table 3 cancers-13-00378-t003:** Outcomes, conclusions, and limitations for recurrent CRLM.

Author	Outcomes (NAC + rLT vs. rLT Alone)	Conclusions	Limitations
Adair [45]	Univariable analysis proportional hazards model OS: *p*= 0.250 DFS: not significant (*p*-value NR)	NAC before repeat resection did not reduce OS or affect DFS. NAC before local treatment could enhance resectability rates.	Long study duration
Adam [46]	Log-rank test survival one-year OS: 70% vs. 70% three-year OS: 45% vs. 53% *p* = 0.86	No significant difference in survival for NAC in third hepatectomy. Higher risk of bleeding and more fragile liver caused by chemotherapy.	Limited population
Brachet [48]	Univariate Cox regression analysis survival HR = 1.034 (0.521–2.051) *p* = 0.923	NAC is not a prognostic factor but might increase survival in repeat hepatectomy. NAC improves resectability of repeat resection.	Limited population Long study duration
Butte [49]	Effective salvage therapy 27.3% vs. 24.0% *p* = 0.7	No significant difference in effectiveness of salvage therapy between NAC + resection and resection alone.	Selection bias
Heise [52]	Univariate analysis DFS *p* = 0.483	No significant difference in disease-free survival for perioperative chemotherapy.	Limited population
Neal [59]	Univariate Cox regression survival analysis OS: HR (95% CI) = 0.951 (0.400–2.261) *p* = 0.910 Cancer-specific survival: HR (95% CI) = 1.033 (0.434–2.455) *p* = 0.942	No significant association in OS between NAC + hepatectomy and hepatectomy alone.	Limited population Selection bias
Neeff [60]	Univariate analysis of survival log-rank test one-year OS: 96.9% vs. 100.0% three-year OS: 59.8% vs. 80.0% five-year OS: 50.4% vs. 40.0% *p* = 0.89	Chemotherapy did not univariately affected long-term outcome in survival.	Small sample size
Pessaux [61]	Univariate analysis log-rank test two-tailed five-year OS: 27% vs. 11% *p* = 0.39	Effective chemotherapy and repeat local treatment is suggested for improved prognosis.	Limited population
Valdimarsson [62]	Relative liver volumes after second procedure 100% vs. 91% *p* = 0.200	Liver volume did not significantly differ between patients with and without chemotherapy	Selection bias
Viganò [63]	Univariate analysis log rank test five-year OS: 61.5% vs. 43.7% *p* = 0.021 Multivariate analysis Cox proportional hazard model HR (95% CI) = 0.529 (0.299–0.934) *p* = 0.028	Higher survival rates for patients with preoperative chemotherapy before re-resection. Response to chemotherapy univariately improved survival, but multivariately the prognostic role was not confirmed.	Selection bias
Wicherts [64]	Univariable analysis five-year OS: 39% vs. 36% *p* = 0.572	No significant Repeat local treatment is more challenging after preoperative chemotherapy due to liver damage.	Limited population

Note: NAC = neoadjuvant chemotherapy; rLT = repeat local treatment; OS = overall survival; DFS = disease-free survival; HR = hazard ratio; NR = not reported; CI = confidence interval.

**Table 4 cancers-13-00378-t004:** Conclusions, recommendations, and limitations for recurrent CRLM

Author	Conclusions and Recommendations	Limitations
Andreou [47]	No negative effect of NAC on surgical results of repeat local treatment.	Selection bias
Hallet [50]	The liver is potentially more friable after chemotherapy before repeat local treatment.	Selection bias Information bias Misclassification bias
Hashimoto [51]	Suggestion of aggressive oncosurgical approach if recurrent CRLM is resectable. Chemotherapy and repeat local treatment are related to increased PFS. Evaluation of responsiveness of chemotherapy affected selection of repeat local treatment.	Limited sample size Long study duration Selection bias
Homayounfar [53]	Intensive chemotherapy protocols could qualify a larger group of patients for repeat local treatment.	Selection bias
Imai, 2018 [54]	Prognostic character of chemotherapy remains unclear. Aggressive oncosurgical approach might be associated with increased survival.	Selection bias
Imai, 2019 [55]	Beneficial outcome for patients with NAC and repeat local treatment.	Historical bias
Ishiguro [56]	Chemotherapy before local treatment could prolong survival for patients with risk factors (extended disease).	Selection bias
Kishi [57]	The OS of NAC + repeat resection and repeat resection alone was comparable.	Selection bias
Matsuoka [58]	Aggressive oncosurgical approach should be performed, considering repeat hepatectomy and systemic chemotherapy. It might improve survival in selected patients.	Selection bias

Note: NAC = neoadjuvant chemotherapy; PFS = progression-free survival; OS = overall survival; oncosurgical approach = consists chemotherapy and local treatment.

## Data Availability

The data presented in this study are available on request from the corresponding author.

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
