# Peer review of "The Role of Neoadjuvant Chemotherapy in Repeat Local Treatment of Recurrent Colorectal Liver Metastases: A Systematic Review and Meta-Analysis"

_cancers, 2021, doi:10.3390/cancers13030378_

Round 1
Reviewer 1 Report
Thank you for the privilege to evaluate your study entitled: The Role of Neoadjuvant Chemotherapy in Repeat Local Treatment of Recurrent Colorectal Liver Metastases: a Systematic Review and Meta-Analysis. The paper is brief and is clearly written. The paper discusses the important issue of justification of preoperative chemotherapy in colorectal liver metastases. I have some remarks which should be addressed before the publication of the manuscript.
Major remarks.
My impression is that the significant study limitation is the long study duration. This meta-analysis accepts patients who had started their treatment more than 20 years ago. This results in a very heterogeneous study population. It is difficult to compare patients who underwent very different treatment protocols. In a significant number of studies, the type of administered chemotherapy was not reported. I have serious doubts concerning the inclusion of these papers.
I believe the section “study limitations” is not satisfactory. This study has extreme selection biases. Please comment on the number of factors influencing OS after treatment like metachronous vs. synchronous metastases, single vs. multiple, single organ vs. many sites, resectable vs. RFA, etc. Most of them were not mentioned and the patients were directly included in the analysis.
Lines 269-279. Please limit the conclusions to the direct take-home message of your paper. Please do not comment on the current guidelines or planned papers. This should be mentioned in the discussion instead of conclusions.
Minor remarks.
Table 1. You mention that only retrospective papers were included. In this case, there is no need for the column: study design. Please delete.
Reviewer 2 Report
The paper is of interest. The systematic review and meta-analysis is well-written.
I think the discussion can be developed a little more. For example, it is necessary to identify the prognosis for these relapsed patient. A normogram for patients with refractory colorectal patients (The colon life normogram, Annals of Oncology, Volume 28, Issue 3, 2017, Pages 555-561) for example estimates this prognosis.
Another important point is the schedule of NAC: with or without associated biological agent? Many different NACs have been used so far, with a huge heterogeneity.
I would add a paragraph of conclusions, apart from discussion.
Probably appendix A with search strategies is redundant and of small interest to the reader.
